# Repurposing Probenecid to Inhibit SARS-CoV-2, Influenza Virus, and Respiratory Syncytial Virus (RSV) Replication

**DOI:** 10.3390/v14030612

**Published:** 2022-03-15

**Authors:** Ralph A. Tripp, David E. Martin

**Affiliations:** 1Department of Infectious Diseases, University of Georgia, Athens, GA 30602, USA; 2TrippBio, Inc., Jacksonville, FL 32256, USA; davidmartin@trippbio.com

**Keywords:** repurposed, antiviral, respiratory virus, influenza virus, SARS-CoV-2, RSV, replication, screens

## Abstract

Viral replication and transmissibility are the principal causes of endemic and pandemic disease threats. There remains a need for broad-spectrum antiviral agents. The most common respiratory viruses are endemic agents such as coronaviruses, respiratory syncytial viruses, and influenza viruses. Although vaccines are available for SARS-CoV-2 and some influenza viruses, there is a paucity of effective antiviral drugs, while for RSV there is no vaccine available, and therapeutic treatments are very limited. We have previously shown that probenecid is safe and effective in limiting influenza A virus replication and SARS-CoV-2 replication, along with strong evidence showing inhibition of RSV replication in vitro and in vivo. This review article will describe the antiviral activity profile of probenecid against these three viruses.

## 1. Introduction

Key steps in drug discovery include target identification and screening. Typically, the viral components are the first choice of target, followed by host factors. Host factor targeting in antiviral drug development generally requires a deeper understanding of the virus–host interface, as well as the mechanisms of action of the antiviral drug, so as to inhibit viral replication. In addition, when evaluating drug inhibition of viral replication, it is important to evaluate the drug dose ranges required for efficacy using translatable human cell lines and in preclinical models of infection to determine the 50% and 90% inhibitory concentrations (IC_50_/IC_90_), as if too high it is likely that the drug may not be useful. Recent drug screening methods have transitioned away from the more laborious virus plaque assay method traditionally used to quantitate viral replication, shifting to measuring inhibition using automated high-throughput assays that allow for screening of hundreds of drugs. These screening methods are based on random screening of potential antiviral drugs using new assay methods to enable the homogeneous measurement of endpoints such as cytopathic effect, viral protein, or reporter gene expression, and assume that they can serve as markers of viral replication [1,2,3,4]; however, they typically do not directly measure or quantitate drug inhibition of viral replication. Classical methods to determine drug efficacy assess a reduction in viral replication using a virus plaque assay, and may measure a decrease in viral RNA transcription as determined by real-time polymerase chain reaction (RT-PCR). The virus plaque assay can be affected by both the viral input (multiplicity of infection (MOI)) and the duration or kinetics of the assay [5]. It is important to understand how variable assay conditions may affect the results in drug screenings, and to understand what the cell-based readouts measure. There are many issues deducing drug screening results using newer methods, such as the reporter cell lines used. Specifically, the reporter cell line often uses a different virus for infection that is supposed to emulate infection with a more serious (BSL3) pathogen. For example, occasionally a replicon with a modified viral subgenome capable of self-replicating without producing infectious virus is used [6]. In addition, the reporter cells used may not be relevant to human disease, or may not fully support viral replication, and/or the reporter cell line may require culturing with media containing serum or pretreating with serum, thereby affecting the activity of the drug compound. While these newer screening methods may be more time-efficient and allow for high throughput, in the end, it is critically important to measure or compare drugs’ screening results to endpoints that directly measure viral replication, e.g., plaque assays.

Viral replication and transmissibility are the principal causes of endemic and pandemic disease threats. There remains a need for broad-spectrum antiviral agents. Viruses use host cell machinery for replication. Most antiviral approaches target the virus to limit potential interference with the host cell function. This strategy can be parochial, as it limits the scope of potential druggable targets. Additionally, features of the virus also limit the scope of antiviral agents due to differences between RNA and DNA viruses, cytoplasmic or nuclear replication, and the degree of reliance on host cell proteins for viral replication. The majority of antiviral agents are highly targeted to a specific virus or viral family. Current tendencies in antiviral drug development are to use structural and systems methods to identify new drug targets shared across specific viral pathogens. This approach is reflected in the Antiviral Drug Discovery (AViDD) Centers for Pathogens of Pandemic Concern funding opportunity announcement (FOA) issued by the NIH for early summer 2022 funding (https://grants.nih.gov/grants/guide/rfa-files/RFA-AI-21-050.html, accessed on 14 March 2022). The goal of these multidisciplinary AViDD research centers is to develop novel antiviral drug candidates with the potential to address SARS-CoV2 as well as to be able to pivot to address future viral outbreaks or pandemics. Thus, most antiviral drugs will aim to block the function of a specific viral protein that may also affect viruses in a family. While there are few broad-spectrum antivirals, there are exceptions, including the influenza virus RNA-dependent RNA polymerase inhibitor favipiravir, which has broad-spectrum RNA viral activity against the influenza virus, Ebola virus, and Lassa fever virus [7,8,9]. In addition, ribavirin, which is a nucleoside analog that inhibits viral polymerase enzymes, has broad-spectrum activity against RNA viruses, including RSV, hepatitis, influenza, parainfluenza viruses, metapneumovirus, and New- and Old-World arenaviruses [10,11,12]. There are other broad-spectrum drugs being evaluated [13,14], but balancing drug efficacy with host cell toxicity remains to be achieved. Alternatively, drug repurposing—involving the use of an existing drug to treat an illness different from what it was originally designed for—has been examined. For example, multiple clinical trials have been performed in an attempt to find drugs to fight COVID-19 [15,16,17], and several drugs have recently received emergency use authorization (EUA) by the FDA to treat mild-to-moderate coronavirus, including Paxlovid, molnupiravir, and remdesivir [18,19,20].

## 2. Respiratory Viruses

Respiratory viruses are the most frequent cause of disease in humans, having significant impacts on morbidity and mortality worldwide [21,22,23]. The most common respiratory viruses are endemic agents such as coronaviruses (CoVs), respiratory syncytial viruses (RSVs), and influenza (flu) viruses. Although vaccines are available for CoV2 and flu, there is a paucity of effective antiviral drugs. For RSV, there is no vaccine available, and therapeutic treatments are very limited [24,25]. Moreover, there are few preventive or therapeutic interventions available for respiratory viruses. A better understanding of the host–virus association is needed in order to develop or repurpose drugs, as all viruses usurp cell pathways for replication. Presently, the world is experiencing a SARS-CoV2 (CoV2) pandemic that primarily ravages unvaccinated populations, particularly as new variants emerge; however, they also may infect vaccine-protected populations [26,27]. It is clear that novel CoV2 strains pose significant threats, and that therapeutic antiviral drugs are needed. CoVs are classified into four genera that include alpha, beta, delta, and gamma CoVs. The prototype human CoVs are OC43, HKU1, 229E, and NL63, which cause common colds and self-limiting upper respiratory infections, while SARS-CoV, CoV-2, and MERS can cause epidemics of varying severity [28]. Frighteningly, RSV and flu may co-circulate with CoV2 (or other CoVs, e.g., Middle East respiratory syndrome (MERS)), where the specter of viral co-infection could cause increased morbidity and mortality. RSV is the most common cause of serious lower respiratory tract infection (LRTI) in young and elderly patients, where most young children experience RSV infection by the age of two, and reinfections are common throughout life [29,30]. RSV strains A and B co-circulate having no significant correlation with disease severity [31,32]. RSV is estimated to cause >3 million hospitalizations and 95,000–150,000 deaths globally each year, where up to 175,000 hospitalizations in the United States are children < 5 years of age [21,33,34]. RSV is also estimated to cause > 14,000 deaths/year in adults in the United States. Juxtaposed, influenza A virus (IAV) or influenza B virus (IBV) may cause substantial infection and disease. Similar to RSV infection, seasonal flu epidemics occur in children and adults, but severe cases generally occur in the very young or old The burden of flu in the United States can vary widely. The CDC estimates that flu resulted in 9–41 million illnesses, 140,000–710,000 hospitalizations, and 12,000–52,000 deaths annually between 2010 and 2020 [35,36,37]. As for CoV2, it is undeniable that therapeutic antiviral drugs are needed, and a safe and effective broad-spectrum antiviral would be an excellent outcome. Recently, EUAs for Paxlovid, molnupiravir, and remdesivir were issued, which should help facilitate antiviral drug therapy for COVID-19 [20,38,39]. Paxlovid is a main protease (Mpro) inhibitor that has been shown to reduce hospitalization or death by 89% when administered within 3 days of symptom onset; molnupiravir incorporates the wrong bases into new viral RNA, and has been shown to reduce hospitalization or death by 30% when administered within 5 days of symptom onset, while remdesivir is a nucleotide prodrug of an adenosine analog that has been shown to reduce time to clinical improvement [40]. The oral antivirals (Paxlovid and molnupiravir) can be used in non-hospitalized patients, raising the hopes for COVID-19 treatments, but there is room for improvement of the drug repertoire, as antiviral drugs are not always efficacious, owing to viral resistance.

## 3. The Virus–Host Cell Interface for Viral Replication

COVID-19 has accentuated the need for drug discovery, and this has helped propel the need for a better understanding of virus–host cell interactions, with the hope that this could facilitate drug breakthroughs [41]. Antiviral drugs can target the virus directly, or the cellular components that are used for viral replication. The direct antiviral strategy is the most common, but is likely to lead to a narrow spectrum of antiviral activity, with a higher likelihood of developing drug resistance. In contrast, a drug that targets a host cell pathway to impede viral replication is more likely to have a broad spectrum and inhibit multiple viruses that require that cellular pathway for replication. Most of the current drugs are small molecules whose competitive binding to proteins makes them specific therapeutic agonists or antagonists. Traditional drug discovery pipelines are time- and resource-intensive, and unable to rapidly respond to outbreaks or pandemics, but by bridging high-throughput screening (HTS) with host gene silencing—i.e., RNA interference (RNAi)—it is possible to rapidly discover the host pathways used by viruses for their replication [42,43]. HTS–RNAi is a powerful tool and provides an unbiased approach to identify essential genes that facilitate viral replication, as well as confer resistance or sensitivity to drug treatment. RNAi-silenced genes associated with a loss (inhibiting viral replication) or gain (increasing viral replication) of function can be rapidly identified. The loss of gene function by RNAi has been used to determine whether drug treatments can reproduce the same effect on viral replication. Several genome-wide RNAi screenings have identified pro- and antiviral host genes that affect viral replication. Genome-wide RNAi screenings have aided in the discovery of host genes and pathways that are co-opted by respiratory viruses [44,45,46,47,48], and these screenings have been used to aid in developing vaccine cell lines [49,50,51,52], as well as to evaluate repurposed drugs for CoV2, flu, and RSV [53,54,55].

As viruses rely on host proteins, targeting these host proteins to inhibit viral replication may prevent drug resistance, leading to a broad-spectrum therapeutic approach, given how viruses exploit common cellular pathways [56]. However, this approach requires a thorough knowledge of virus–host interactions and their biological significance for viral replication. Studies exploring RNAi screenings of human type II A549 cells infected with A/WSN/33 and other flu strains revealed key host genes needed for flu replication [55]. One of the host genes identified was the organic anion transporter-3 gene (OAT3)—a member of the solute carrier (SLC) superfamily, which comprises 298 members grouped into 43 families, including SLC22A8, and the SLC22 family is subdivided into organic cation transporters, zwitterion/cation transporters, and organic anion transporters (OATs) [57,58,59]. Transfection of A549 cells with siRNA targeting the OAT3/*SLC22A8* gene silenced flu replication [55]. As OAT3 is important for flu replication, a drug that inhibits OAT3 should be effective as an antiviral drug for flu. Studies showed that probenecid—a classical clinical inhibitor of OAT1 and OAT3—reduces OAT3 mRNA and protein levels, and probenecid treatment in vitro and in vivo reduced flu lung titers in a murine model [55].

## 4. Probenecid as an Antiviral

Probenecid is an FDA-approved drug with a well-documented (>7 decades) safety profile for treating gout and hypertension, is a nonspecific inhibitor of OATs that blocks transport by OAT1, OAT3, and URAT1, and has been used to extend the plasma half-life of β-lactam antibiotics and oseltamivir [60,61,62,63,64]. Published reports have shown that probenecid prophylaxis or treatment can inhibit IAV (WSN/33, A/New Caledonia/20/99, A/California/07/09, and A/Philippines/2/82/X-79) replication in A549 cells and in BALB/c mice [55]. Specifically, it was shown that probenecid is effective in limiting flu replication in vitro at an IC_50_ ranging between 5 × 10^−5^ and 5 × 10^−4^ μM in A549 cells. Mice treated prophylactically with 200 mg/kg of probenecid (24 h pre-infection), therapeutically with 200 mg/kg probenecid (24 h post-infection), or administered 25 mg/kg probenecid daily for 3 days following infection had reduced morbidity and mortality. 

CoVs, like flu, require host cell components to replicate. Recently, it was shown that probenecid prophylaxis or treatment inhibited CoV2 (hCoV-19/USA/CA_CDC_5574/2020, or the variant of concern hCoV-19/USA/CA_CDC_5574/2020, B.1.1.7) replication in Vero E6 cells, normal human bronchial epithelial (NHBE) cells, and CoV2-infected Syrian hamsters [54]. Specifically, probenecid treatment blocked CoV2 replication in cells treated with nanomolar concentrations of probenecid (0.00001–100 μM), as well as in hamsters prophylactically or therapeutically treated with probenecid, reducing viral lung titers by 4–5 logs compared to the controls, which were approximately 10^9^ logs of the virus.

Consistent with the findings for flu and CoV2, a recent study showed that probenecid treatment reduced the replication of three strains of RSV—i.e., strains A2, Memphis-37, and B1—in human respiratory epithelial cell lines as well as in BALB/c mice (preprint: DOI:10.21203/rs.3.rs-1280404/v1, https://assets.researchsquare.com/files/rs-1280404/v1/4ecb646c-cebd-41b0-85f1-0fccb802ba74.pdf?c=1643732484; accessed on 14 March 2022). 

In this study, probenecid treatment of Vero E6 cells, HEp-2 cells, or NHBE cells was very effective at preventing RSV replication. In mice, probenecid administered prophylactically before RSV infection was shown to reduce viral lung titers, and probenecid given therapeutically also reduced viral lung titers, demonstrating the versatility of probenecid as a chemotherapeutic. 

Efforts to translate probenecid as an antiviral drug in preclinical studies to clinical practice are ongoing, with clinical trials being planned for patients infected with CoV2, influenza, and RSV.

## 5. Probenecid and Inflammation

Probenecid can modulate the expression of ACE2 [65], and targets the pannexin-1 gene, *PANX1* [66]. *PANX1* is an ATP release channel that facilitates the communication between T cells to inhibit the severity of airway inflammation [67,68]. *PANX1* limits airway inflammation driven by type-2 CD4+ T-cell inflammatory responses, and probenecid is an inhibitor of *PANX1* [69] that mediates the activation of caspase-1 and release of IL-1β induced by P2X7 receptor activation [70]. *P2X7R* is expressed in macrophages and other immune cells, and is an ion channel gated by high concentrations of extracellular ATP, i.e., concentrations known to be present at sites of inflammation [71,72]. Interestingly, CoVs and RSV replicate in the host cell’s cytosol, while flu replicates in the nucleus; however, all of these viruses are susceptible to probenecid. As viruses modify host genes and pathways to facilitate viral replication, it is possible that probenecid treatment may modify viral-mediated changes that contribute to the treatment of “long COVID” symptoms [73,74], where *PANX1* expression in the brain and synaptic plasticity are modified by COVID [75,76].

## 6. Probenecid Pharmacokinetics

A key consideration when repurposing a drug for a new indication is the ability of the drug to achieve pharmacologically relevant plasma concentrations against the new drug target for an appropriate duration of exposure. Although a drug may exhibit in vitro potency against a new target, the effective concentration range is not always within the clinically relevant plasma concentrations. A recent example of this is ivermectin [77]. The reported in vitro activity for ivermectin against CoV2 was 2 µM; however, a pharmacokinetic/pharmacodynamic (PK/PD) analysis [78] of available ivermectin pharmacokinetic data found that the reported IC_50_ was >9-fold higher than achievable plasma concentrations with a standard 600 μg/kg dose daily for 3 days, and >21-fold higher than the expected lung tissue concentration following the same dose. Pharmacologically, ivermectin would not be expected to exert an antiviral effect against Cov2.

In a recently published PK/PD analysis [54], a population pharmacokinetics (pop-PK) model was developed to characterize probenecid PK using a one-compartment structure with saturable elimination and first-order absorption. Simulations using the final pop-PK model to generate probenecid exposure profiles comparing the administration of 600 mg twice daily, 900 mg twice daily, or 1800 mg once daily were completed, and free drug concentrations were calculated (Table 1).

The data in Table 2 show the IC_90_ values for CoV2 [54], flu [55], and RSV (preprint: DOI:10.21203/rs.3.rs-1280404/v1) versus the predicted trough-free drug concentrations for the 900 mg BID dosing regimen for probenecid at a steady state. These data suggest that this dosing regimen would result in probenecid exposures that are >4-fold for RSV and >50-fold for CoV2. Furthermore, based on these results, it is likely that a much lower dose of probenecid could be effective for influenza, as the ratio of the free trough concentration is estimated to be >33,000-fold higher than the IC_50_ value.

## 7. Conclusions

Respiratory viruses are frequent causes of disease in humans. Some of the more common respiratory viruses are endemic agents such as CoVs, flu, and RSV. These viruses all consist of variants or strains. For example, human CoVs include strains OC43, HKU1, 229E, and NL63 that cause common colds, while CoV2 can cause pandemics, and RSV strains A and B co-circulate, similarly to various influenza A virus (IAV) or influenza B virus (IBV) strains. Thus, the ideal antiviral drug would have a broad spectrum and inhibit replication of these common respiratory viruses, be non-cytotoxic, and reduce inflammation associated with the infection process. Preclinical studies have shown that probenecid meets or exceeds these criteria, having no detectable cytotoxicity, yet inhibits virus replication in a dose-dependent fashion at nanomolar-to-micromolar concentrations [54,55]. Recent data have shown that probenecid has broadly potent in vitro antiviral activity against CoV2 (WA1/2020) and variants B.1.1.7 (Alpha), B.1.617.2 (Delta), and B.1.1.529 (Omicron), as well as RSV strains A and B. Furthermore, this activity has been confirmed in in vivo animal studies for each virus, where significant reductions in viral lung titers were observed. 

The antiviral mechanism of action appears to be linked to the solute carrier (SLC) superfamily (e.g., *SLC22A8*) and their function as organic cation transporters (OATs). OATs are important for anion exchange and facilitate peptide transport—a feature that has been co-opted by viruses and used for the assembly of virions. Probenecid is a chemical inhibitor of OATs, including *OAT3*. Additionally, probenecid can modulate the expression of ACE2 and target the pannexin-1 gene. Pannexin-1 mediates the activation of caspase-1 and release of IL-1β, affecting inflammation, so probenecid may have additional anti-inflammatory activity. The anti-inflammatory effect of probenecid associated with *PANX1* may provide a clinical benefit.”.. Finally, predicted human plasma concentrations are expected to greatly exceed the protein-binding-adjusted IC_90_ value across the entire dosing interval, giving probenecid an ideal profile as an antiviral agent. The potent antiviral and anti-inflammatory activity along with the favorable clinical pharmacology strongly support the continued clinical investigation of probenecid as a potential therapeutic or prophylactic option for CoV2 (COVID-19), flu, or RSV, and clinical trials are currently being planned.

## Figures and Tables

**Table 1 viruses-14-00612-t001:** Probenecid steady-state concentration and free drug concentrations after different probenecid doses.

Dose (mg)	Frequency	Cmax_ss_ (µg/mL)	Cmin_ss_ (µg/mL)	Free Cmax_ss_ (µg/mL)	Free Cmin_ss_ (µg/mL)
600	BID	63	33	3.1	1.7
900	BID	142	95	7.1	4.8
1800	QD	180	65	9.0	3.3

Cmax_ss_ = steady-state maximum concentration; Cmin_ss_ = steady-state minimum concentration; Free Cmax_ss_ = steady-state maximum concentration adjusted for 95% protein binding; Free Cmin_ss_ = steady-state minimum concentration adjusted for 95% protein binding; BID = twice a day; QD = once a day.

**Table 2 viruses-14-00612-t002:** Probenecid free drug concentrations at trough and target IC_90_ values for CoV2, flu, and RSV.

	CoV2 [54] (NHBE Cells)	Influenza A [55] (A549 Cells)	RSV Treatment ^ (NHBE Cells)	RSV Prophylaxis ^ (NHBE Cells)
IC_90_ (µM)	0.3	0.0005	2.7	3.6
IC_90_ (µg/mL)	0.086	0.00014268	0.8	1.0
Free Cmin_ss_ at 900 mg BID (µg/mL)	4.8	4.8	4.8	4.8
Ratio trough/IC_90_	56	33,649	6.0	4.8

^ DOI:10.21203/rs.3.rs-1280404/v1; COV2 = hCoV-19/USA/CA_CDC_5574/2020; Flu = A/WSN/33; RSV = strain A2.

## Data Availability

Not applicable.

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
