# Peer review of "Repurposing Probenecid to Inhibit SARS-CoV-2, Influenza Virus, and Respiratory Syncytial Virus (RSV) Replication"

_viruses, 2022, doi:10.3390/v14030612_

Round 1

Reviewer 1 Report

In this paper drug repurposing as a method to identify novel antiviral therapies for respiratory viruses is reviewed, with focus on the drug probenecid. Probenecid is a drug which is currently used to treat gout, but it also has anti-inflammatory properties. A recent study in cell lines and hamsters showed that probenecid resulted in inhibition of SARS-CoV-2 replication. Similar data were also obtained with probenecid in Influenza Virus, and Respiratory Syncytial Virus replication, both in vitro and in vivo. The part in the review about Probenecid is novel, however there is a lot of overlap with the primary published articles of Probenecid in SARS-CoV-2, and Influenza Virus.

Main comments:

  1. The authors are employees of TrippBio, a company created to develop and commercialize broad spectrum anti-viral drugs. A conflict-of-interest statement needs to be added.
  2. How do the results of other anti-viral drugs with activity against CoV2 compare with the data obtained with probenecid?
  3. The translation to clinical practice needs to be addressed. What will be the next step? Have clinical studies started already?
  4. Several potential mechanisms of action are mentioned, including OAT3 inhibition, modulation of ACE2 expression, and PANX1 inhibition. Where do the authors think the main activity comes from?

Reviewer 2 Report

The manuscript collects recent findings on the antiviral properties of probenecid against notable respyratory viruses like influenza virus, SARS-CoV2 and RSV.

The manuscript is well organised, each section has been properly developed.

English style is accurate, references are adequate. I also suggest citing and describing the following manuscripts:

In the paragraph “the virus-host cell interface…”

Tonelli M, Naesens L, Gazzarrini S, Santucci M, Cichero E, Tasso B, Moroni A, Costi MP, Loddo R. Host dihydrofolate reductase (DHFR)-directed cycloguanil analogues endowed with activity against influenza virus and respiratory syncytial virus. Eur J Med Chem. 2017 Jul 28;135:467-478. doi: 10.1016/j.ejmech.2017.04.070.

In the paragraph “probenecid and inflammation”

Rosli S, Kirby FJ, Lawlor KE, Rainczuk K, Drummond GR, Mansell A, Tate MD. Repurposing drugs targeting the P2X7 receptor to limit hyperinflammation and disease during influenza virus infection. Br J Pharmacol. 2019 Oct;176(19):3834-3844. doi: 10.1111/bph.14787.

Luu R, Valdebenito S, Scemes E, Cibelli A, Spray DC, Rovegno M, Tichauer J, Cottignies-Calamarte A, Rosenberg A, Capron C, Belouzard S, Dubuisson J, Annane D, de la Grandmaison GL, Cramer-Bordé E, Bomsel M, Eugenin E. Pannexin-1 channel opening is critical for COVID-19 pathogenesis. iScience. 2021 Dec 17;24(12):103478. doi: 10.1016/j.isci.2021.103478.

The conclusion of the paper is too short and generic; the Authors' own thoughts and perspectives should be better pointed out on the antiviral (through host factors inhibition) and anti-inflammatory properties of probenecid. What can be the main determinant of probenecid for an effective inhibition of virus replication? Can probenecid be considered a broad spectrum antiviral agent? What can we derive for future research and development?

I consider this manuscript of interest for readers working in the field, and I suggest its publication after having addressed the indicated revisions.
